# 3D Printed Models—A Useful Tool in Endovascular Treatment of Intracranial Aneurysms

**DOI:** 10.3390/brainsci11050598

**Published:** 2021-05-06

**Authors:** Emilia Adriana Marciuc, Bogdan Ionut Dobrovat, Roxana Mihaela Popescu, Nicolaie Dobrin, Alexandru Chiriac, Daniel Marciuc, Lucian Eva, Danisia Haba

**Affiliations:** 1Department of Radiology, University of Medicine and Pharmacy “Grigore T. Popa”, 700115 Iasi, Romania; emma.marciuc@gmail.com (E.A.M.); drpopescuroxana@yahoo.com (R.M.P.); danihaba@yahoo.com (D.H.); 2Department of Neurosurgery, Emergency Hospital “Prof. Dr. N. Oblu”, 700309 Iasi, Romania; dobrin_nicolaie@yahoo.com (N.D.); chiriac_a@hotmail.com (A.C.); elucian73@yahoo.com (L.E.); 3Department of Oral and Maxillofacial Surgery, University of Medicine and Pharmacy “Grigore T. Popa”, 700115 Iasi, Romania; d.marciuc@yahoo.com

**Keywords:** 3D printing, intracranial aneurysm, endovascular approach, coil embolization

## Abstract

Many developments were made in the area of endovascular treatment of intracranial aneurysms, but this procedure also requires a good assessment of vascular anatomy prior to intervention. Seventy-six cases with brain aneurysms were selected and 1:1 scale 3D printed models were created. We asked three interventional neurosurgeons with different degrees of experience (ten years, four years, and a fourth-year resident) to review the cases using CTA (computed tomography angiogram) with MPR (multiplanar reconstructions) and VRT (volume rendering technique) and make a decision: coil embolization or stent-assisted coil embolization. After we provided them with the 3D printed models, they were asked to review their treatment plan. Statistical analysis was performed and the endovascular approach changed in 11.84% of cases for ten-year experienced neurosurgeons, 13.15% for four years experienced neurosurgeon, and 21.05% for residents. The interobserver agreement was very good between the ten years experienced interventionist and four years experienced interventionist when they analyzed the data set that included the 3D printed model. The agreement was higher between all physicians after they examined the printed model. 3D patient-specific printed models may be useful in choosing between two different endovascular techniques and also help the residents to better understand the vascular anatomy and the overall procedure.

## 1. Introduction

Intracranial aneurysms are vascular lesions that are often asymptomatic, but when rupture occurs, the consequences can be devastating. Subarachnoid hemorrhage is the most common complication and is also responsible for a mortality rate of 45% and high morbidity among survivors [1]. After ISAT study publication [2] and an increased number of patients treated endovascularly, there were many developments regarding this treatment technique [3,4], but still, it requires a very good understanding of the patient’s specific vascular anatomy.

Despite the advances made in imaging and postprocessing tools with multiplanar reconstructions and volume rendering techniques that enable a better overall understanding of the aneurysms [5,6], three dimensional (3D) technology provides another perspective with a tangible 3D printed model, neurosurgeons being able to visualize an aneurysm beyond a flat-screen.

Since the early 2000’s, this technology has been used in the medical field, especially in hard tissue applications, but with the development of different 3D printers and materials, brain vascular lesions like intracerebral aneurysms can now be reproduced accurately.

3D printing of intracranial aneurysms represents a field of interest that is nowadays investigated through many studies due to their potential to help neurosurgeons or interventional radiologists choose a more suitable approach.

The aim of this study is to evaluate the role of 3D patient-specific printed models in establishing the endovascular approach (coil embolization or stent-assisted coil embolization) in 76 cerebral aneurysms and also the agreement among more or less experienced interventional neurosurgeons.

## 2. Materials and Methods

### 2.1. Study Population

This is a retrospective study of patients admitted to Emergency Hospital “Prof. Dr. N. Oblu” Iasi, Romania, a center specialized in neurology/neurosurgery pathologies, between April 2019 to March 2020 who have undergone aneurysm embolization and had a CTA examination in our department of Radiology. We excluded children, patients with multiple aneurysms, and with aneurysms that were in close proximity to bone structures. The final sample included a number of 76 patients. All endovascular procedures were done using an Infinix-I biplane system by Canon. The 3D printing of models included a specific workflow from acquisition imaging data, postprocessing with vascular segmentation to the physical creation of a solid model using a resin printing device.

### 2.2. Acquisition and Postprocessing of CTA Data

All included patients had CTA examination performed by a 32-slice CT scanner (Aquilion Lightning, Canon Medical System, Tustin, CA, USA). The scanning parameters were a: scan range, after a lateral scout view, from the C1 and C2 level to vertex; gantry rotation time, 0.75 s; slice thickness, 0.5 mm; field of view (FOV), 240 mm; tube potential, 110 kV, and tube current, 300 mA. A total amount of 80–90 mL of intravenous contrast media was injected at a flow rate of 4 mL/s. CT angiography data were achieved in caudocranial orientation with no angulation of gantry.

Images were exported in picture archiving and communication system (PACS) in standard DICOM format (digital imaging and communication in medicine). The data was loaded to the segmentation software, 3D Slicer, a free and open-source application for medical image manipulation [7]. With this software, the aneurysm and the parent vessels were segmented using a threshold method that selected a range of pixels between a predefined interval related to the density of enhanced vessels. The segmentation was done by radiologists, and the non-important structures were removed (Figure 1). The final output was exported to an STL file (STereoLithography) that could be read by the printer using proprietary software, Z-Suite (Zortrax S.A., Olsztyn, Poland) [8] (Figure 2).

### 2.3. Printing of 3D Models

For creating real-life-sized models, we used a Zortrax Inkspire printer (Zortrax S.A., Olsztyn, Poland) with a technology that relies on a high-resolution LCD screen with UV LED backlighting that can cure photopolymers layer by layer with a total printing volume of 132 × 74 × 175 mm. The resolution on XY axes and the layer thickness was 50 microns. The material we used was Zortrax Transparent Yellow Flexible Resin. After the printing was completed, the models were immersed in an ultrasound bath with isopropyl alcohol 99%, and the supports were removed (Figure 3).

### 2.4. Case Analyses

Three neurosurgeons, members of our vascular team, MD1 with over 10 years of experience, MD2 with 4 years of experience, and MD3, a fourth-year resident, were asked to review the anonymized cases having access to all the clinical data and all the imaging studies that included NECT (non-enhanced CT) and CTA with MPR and VRT reconstruction. They noted the type of endovascular procedure they would have chosen (direct coil embolization or stent-assisted coil embolization), taking into consideration the shape, size, neck, and orientation of the aneurysm, the parent vessel, and whether there is an emerging vessel from the aneurysmal sac. After that, we provided them with the 3D printed models, and they were asked to review their treatment plan.

### 2.5. Statistical Analysis

We measured the degree of interobserver agreement calculating the Cohen weighted-kappa statistics. Kappa coefficients were determined for the decision that interventionist surgeons have taken before and after using the 3D models in their study analyses. Kappa values lie in a range of 0 to 1. Values between 0.8–1 represented a very good agreement or almost perfect agreement; good agreement or substantial agreement between 0.6–0.8; moderate agreement between 0.4–0.6; poor or fair agreement between 0.2–0.4; slight agreement for values less than 0.2; for values below 0, no agreement [9].

Statistical significance was established at a *p*-value below 0.05. Statistical analysis was performed using SPSS (IBM SPSS Statistics, release 24.0, Armonk, New York, NY, USA).

## 3. Results

The patients in our study were aged from 19 to 75 years (median 54 years) and included 43 women and 33 men. The most common findings were ICA (internal carotid artery) aneurysms (30%) followed by AcoA (anterior communicating artery) aneurysms (26%), MCA (middle cerebral artery) (23%), and posterior circulation aneurysms (21%) (PICA: posteroinferior cerebellar artery; PCA: posterior cerebral artery; AICA: anteroinferior cerebellar artery).

When the observers analyzed the cases without a 3D printed model, the inter-observer agreement was good between MD1 and MD2 (k = 0.703) and between MD1 and MD3 (k = 0.64) (Table 1). After analyzing the cases with a 3D printed model, the kappa values were k = 0.85 between MD1 and MD2, k = 0.70 between MD1 and MD3, and k = 0.72 between MD2 and MD3 (Table 2).

After observers evaluated the 3D printed models, MD1 changed the initial decision in 11.84% of cases, MD2 in 13.15% of cases, and MD3 in 21.05% of cases (Table 3).

## 4. Discussion

Medical 3D printing has been reviewed in many reports since the early 2000’s [10], and it is still being developed because the model accuracy derived from clinical images has been demonstrated by numerous studies published in the last two decades [11,12]. 3D printed models now occupy an important role mainly because of surgeons that saw a high potential in improving their understanding of patients’ specific anatomy and their surgical techniques.

In neurosurgery, the preoperative benefits of 3D printed models of intracranial aneurysms have been demonstrated in many reports [12,13,14] and also their role in surgical training and patient education [15,16,17]. The physical models provide an overview of the complete anatomy of the aneurysm, the relationship with the arterial branching pattern, aspects that were found to be accurate in comparison to the intraoperative reality. Furthermore, they create a good training environment for surgeons to test surgical approaches for clip selection and direction, minimizing the risk of clip malpositioning. Scerrati et al. [15] found a high correspondence between the clipping strategy selected in preoperative planning and the strategy used during surgery. Yin Kang et al. [16], in their study, showed that 3D printed models could also provide the surgeon with good tactile feedback and will help in adjusting the head position and choosing a shorter path to the aneurysm.

The articles published in the literature are focused mostly on surgical or surgical vs. endovascular treatment and simulation and do not include a large sample of printed models. Our study summarizes the workflow to generate 3D models and points out some key elements for speeding the process while maintaining the accuracy of models and also demonstrates the advantages of having a solid 3D model before choosing between two different endovascular procedures for both experienced and less-experienced interventional neurosurgeons.

Endovascular treatment of intracranial aneurysms requires a good assessment of the vascular anatomy and the morphology of the aneurysms in order to choose the treatment approach using metal coils with/without a complementary stent. So, 1:1 scale patient-specific anatomical models play an important role in treatment planning, and the accuracy can be influenced by many variables like image acquisition, postprocessing, and 3D printing process.

In our study, we used 0.5 mm thickness images, and the postprocessing was made by radiologists as suggested by Akmal et al. [18] and Huotilainen et al. [19], ensuring that the interpretation of the images matches the 3D printed model and the structures are being well-differentiated from imaging modality artifacts. For postprocessing, we used 3D Slicer, an open-source software [7] that has been found to be suitable for segmentation as well as in other studies with 3D printing [15,20] because it is free and certified for medical use.

The models were printed using a UV-LCD printer with a higher quality resolution and a good surface finishing (Figure 4) compared to FDM (Fused Deposition Modelling) technology mainly used in other aneurysm printing studies [15,20]. We also had a very good time interval (a mean of 3 h) from data acquisition to the completed model compared with Błaszczyk et al. who disclosed a time-lapse of approximately 4 h [21] or Faraj et al. who disclosed a time-lapse of 24 to 28 h [14].

Studies have demonstrated the anatomical accuracy of the 3D printed models regarding the orientation, diameter, neck of the aneurysm and parent vessels in a qualitative but also quantitative method and even on more sophisticated models like arteriovenous malformations [22,23,24], suggesting that these models can be safely used for preoperative planning.

Analyzing 3D printed models of intracranial aneurysms can make a difference in treatment planning, as we showed in our study. An overview of the anatomy of the aneurysm in a 3D perspective, including the size, orientation, and branching arterial pattern, can help an interventional physician with over 10 years of experience as well as a resident doctor. In our study, we demonstrated a very good agreement (k = 0.85) between the interventionist with 10 years of experience and an interventionist with 4 years of experience when they analyzed the data set that included the 3D printed model (Figure 5), with a kappa coefficient higher than before seeing the solid model (k = 0.7). We also found a good agreement between the surgeon with 10 years experience and the resident (k = 0.7) and between the surgeon with 4 years of experience and the resident (k = 0.724) after they examined the printed model in comparison with before adding the 3D printed model to the data set (k = 0.640 and k = 0.486, respectively). We found no significant discrepancies in our study.

Although computer-assisted 3D morphology assessments are of much value for aneurysm size and neck measurements compared to 2D images [25], a solid 3D model can provide supplementary information, especially in preoperative planning [26,27]. Wang et al. concluded that the printed models were useful for understanding the aneurysm’s structure and for choosing the necessary clips before surgery, mainly for junior neurosurgeons [17], and Hoffman et al. suggested that fabrication of 3D printing and hollow models are helpful in EVAR planning for complex abdominal aorta aneurysms [28]. Tam et al. conducted a study on six different aortic aneurysm cases with complicated anatomical features and investigated the impact of 3D printed models in surgical planning by asking 28 endovascular operators to make a management plan before and after analyzing the model. The treatment plan changed in 20% of cases. The level of confidence also increased to 43% [29]. In our study, the management changed in 11.84% of cases for neurosurgeons with 10-year experience, in 13.15% of cases for neurosurgeons with 4-year experience, and 21.05% of cases for residents. These results suggest that junior neurosurgeons benefit the most from the printed models by having a more comprehensive perspective over the anatomy of the aneurysm.

Our study had several limitations. We did not include in our study internal carotid artery aneurysms that were closely related to the bone structures because we did not perfect a segmentation technique in order to have an accurate 3D printed model that included only the arterial segment without pixels from the bone. Although bone and contrast agent that fills the arterial lumen has different densities, the threshold we used was not sensitive enough to differentiate them in a semi-automatic way, so manual segmentation in these cases was needed.

This study involved statistical data about the interobserver agreement and did not have a “gold standard” reference point regarding the endovascular procedure. We relied on the decision made by an experienced interventional surgeon during the 3D Rotational Angiography, an imaging modality that seems to be becoming the new “gold standard” of intracranial aneurysms diagnosis and characterization [30].

A direction of future research of our work involves the fabrication of hollow models for actual simulation of endovascular procedures and training. We tried to make hollow models for this purpose (Figure 6), but we need to further investigate the postprocessing part of these printed models because they were relatively breakable after three days. So the short period of time may not give a chance for interventional surgeons to make an endovascular simulation. One explanation would be that we did not use the right resin for fabrication because a hollow model must not have porosities and should be relatively flexible. Another reason was probably the fact that we did not expose the model after printing to UV light for curing purposes [31]. However, our classic whole 3D printed models remained intact even after one year.

## 5. Conclusions

Patient-specific 3D printed models can be useful in understanding the morphology of the aneurysms and the surrounding arterial branching pattern and, therefore, can help the interventional team to reach common ground regarding the treatment plan before the endovascular approach. A notable benefit was for junior neurosurgeons that were able to have a different and better perspective of the vessel’s anatomy and of the overall procedure. New research directions include fabricating hollow models with very good material properties that can be used for surgical simulation also imitating the hemodynamics of the aneurysms.

## Figures and Tables

**Figure 1 brainsci-11-00598-f001:**
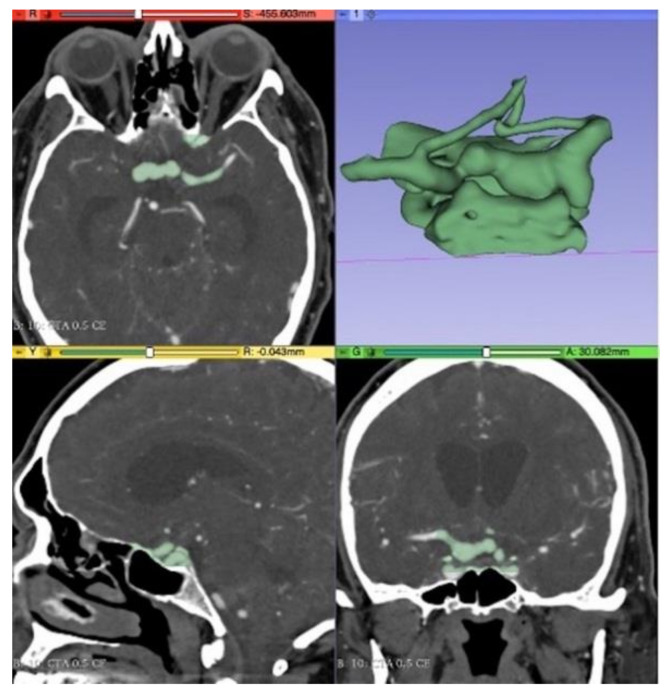
Segmentation using 3D Slicer software.

**Figure 2 brainsci-11-00598-f002:**
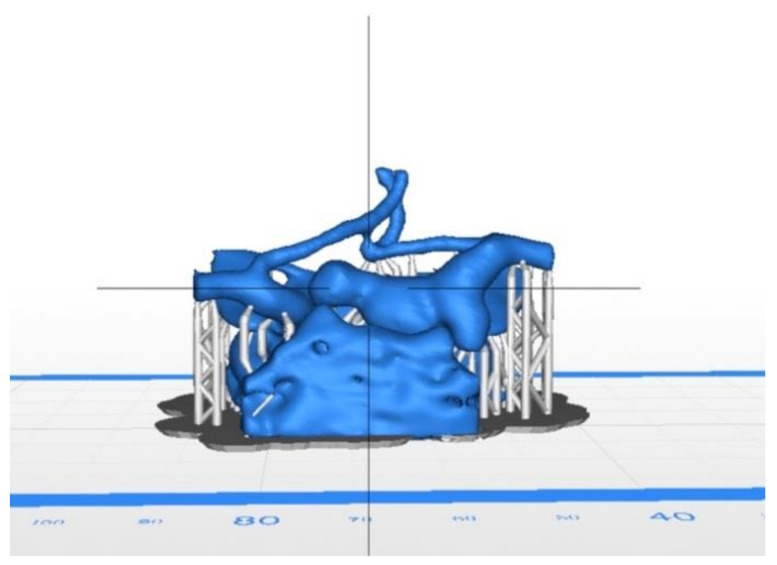
STL format in Z-Suite software.

**Figure 3 brainsci-11-00598-f003:**
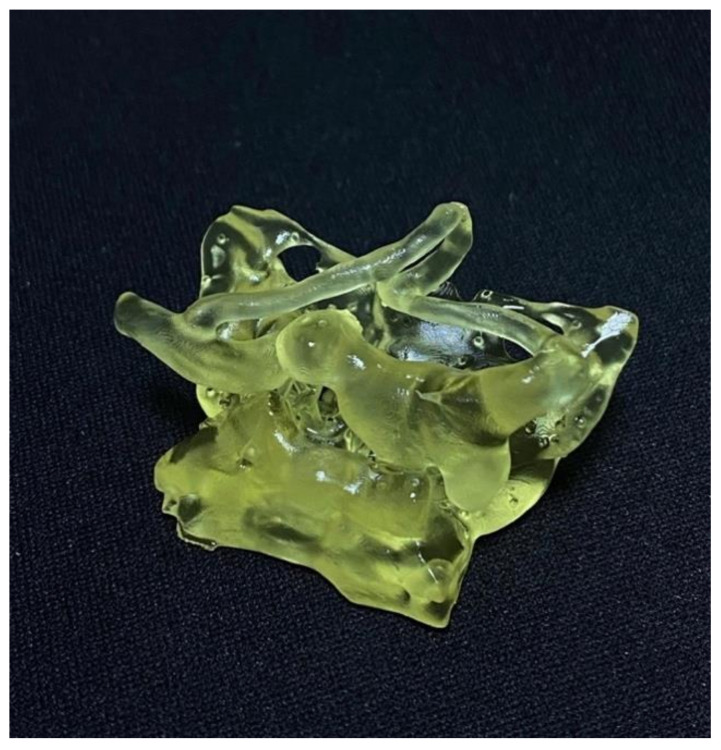
Final 3D printed model of a complex aneurysm of Posterior Communicating Artery.

**Figure 4 brainsci-11-00598-f004:**
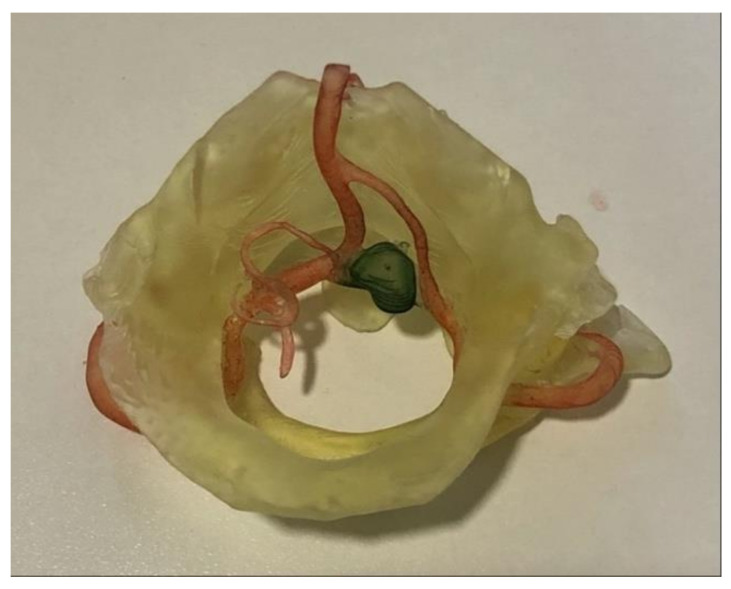
3D printed model of a PICA aneurysm.

**Figure 5 brainsci-11-00598-f005:**
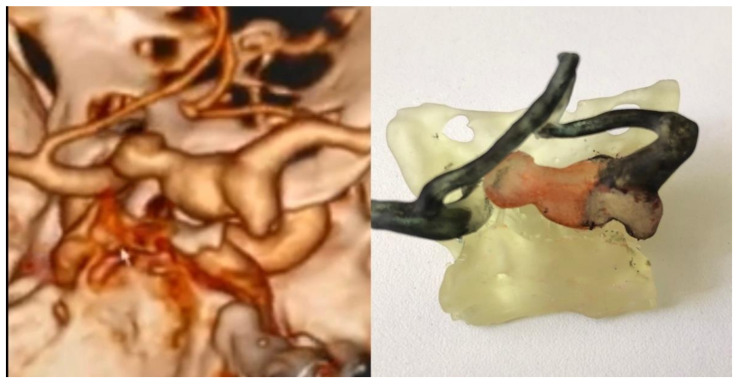
VRT reconstruction from CTA and 3D printed model with a complex aneurysm of Posterior Communicating Artery.

**Figure 6 brainsci-11-00598-f006:**
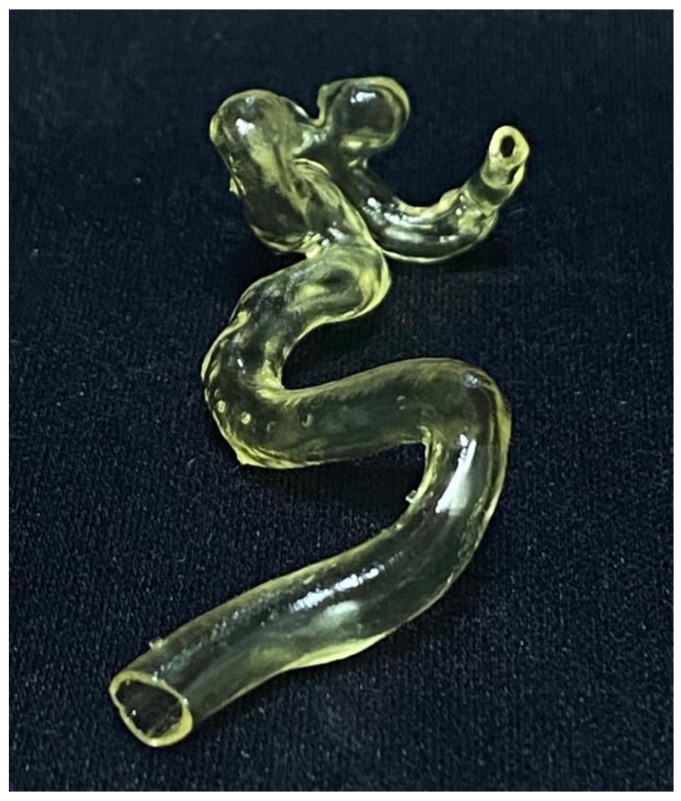
Hollow 3D printed model of ICA with an aneurysm on an ophthalmic artery.

**Table 1 brainsci-11-00598-t001:** Interclass Correlation Coefficient between the three observers before analyzing the 3D printed model.

	MD2	MD3
**MD1**	0.703	0.640
**MD2**	-	0.486

**Table 2 brainsci-11-00598-t002:** Interclass Correlation Coefficient between the three observers after analyzing the 3D printed model.

	MD2	MD3
**MD1**	0.852	0.700
**MD2**	-	0.724

**Table 3 brainsci-11-00598-t003:** Management decision changes for all three observers after analyzing the 3D printed model.

Aneurysm Location		MD1—Decision Changes	MD2—Decision Changes	MD3—Decision Changes
**MCA**	22	2	4	5
**AcoA**	20	1	3	4
**ICA**	18	4	2	3
**Other**	16	2	1	4
**Total decision changes**		9	10	16
		11.84%	13.15%	21.05%

## Data Availability

Data are available upon request; please contact Bogdan Ionut Dobrovat (bogdan.dobrovat@yahoo.com).

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
