# Peer review of "3D Printed Models—A Useful Tool in Endovascular Treatment of Intracranial Aneurysms"

_brainsci, 2021, doi:10.3390/brainsci11050598_

Round 1
Reviewer 1 Report
This topic is interesting, please look at these points:
- Lines 13-14: " Endovascular treatment of intracranial aneurysms is often used nowadays but it’s challenging and there is no unanimity regarding the best approach " This first sentence in the abstract is not correct conceptually and grammatically. Please revise.
- Lines 47-48: "Nevertheless, there are not numerous studies that involve a large number of patients so further in-depth analysis would be useful." What do you mean ? revise this sentence
- Lines 51-52: "between experienced and less experienced interventional neurosurgeons", it is better to write "among more or less experienced interventional neurosurgeons"
- Lines 14-149: "This article discusses the workflow to generate rapid 3D models ... non-experienced interventional neurosurgeons" Improved this section. It is better to remove "From our knowledge, this study includes the highest number of cases (lines 150-151)" what does this sentence apport to the paper?
- Lines 152-156: "Endovascular treatment of intracranial aneurysms requires a good assessment of the... 3D printing process". At this point can be useful to report and discuss the role of 3D printed models also in surgery to help the surgeon to avoid clips malpositioning, wrapping and revascularization. Please look at these references: Revascularization for complex intracranial aneurysms. Neurosurg Focus. 2008;24(2):E21. doi: 10.3171/FOC.2008.25.2.E21. -- Wrapping of intracranial aneurysms: Single-center series and systematic review of the literature. Br J Neurosurg. 2015;29(6):785-91. doi: 10.3109/02688697.2015.1071320. -- Comparison of 3D intraoperative digital subtraction angiography and intraoperative indocyanine green video angiography during intracranial aneurysm surgery. J Neurosurg. 2018 Jul 13;131(1):64-71.
- Lines 182-156: ". In our study, we demonstrated a very good agreement between the 10 years experienced interventionist and 4 years experienced interventionist when they analyzed the data set that included the 3D printed model (Figure 6)" . add percentage data. - "Also, the agreement was higher between all physicians after they examined the printed model." All? How many physicians? add percentage data or removed.
- Lines 220-221: "because they were relatively breakable after 3 days" why ? please discuss
- Figure 4 is useless in this way. Could the authors add more informations?
Author Response
Response to Reviewer 1 Comments
Point 1: Lines 13-14: " Endovascular treatment of intracranial aneurysms is often used nowadays but it’s challenging and there is no unanimity regarding the best approach " This first sentence in the abstract is not correct conceptually and grammatically. Please revise.
Response 1: Lines 13-15 - We replaced the sentence with “Many developments were made in the area of endovascular treatment of intracranial aneurysms but this procedure requires also a good assessment of vascular anatomy prior to intervention.”
Point 2: Lines 47-48: "Nevertheless, there are not numerous studies that involve a large number of patients so further in-depth analysis would be useful." What do you mean ? revise this sentence
Response 2: Lines 51-53 - In this sentence we were referring to the fact that 3D printing in medical field is an area of interest with papers in the literature that study the usefulness of 3D printed models for treatment of brain aneurysms but on small samples of cases. Following your advice, we revised the sentence.
“3D printing of intracranial aneurysms represents a field of interest that is nowadays investigated through many studies due to their potential to help neurosurgeons or interventional radiologists choose a more suitable approach.”
Point 3: Lines 51-52: "between experienced and less experienced interventional neurosurgeons", it is better to write "among more or less experienced interventional neurosurgeons"
Response 3: Lines 56-57 - revised as requested : “… and also the agreement among more or less experienced interventional neurosurgeons.”
Point 4: Lines 14-149: "This article discusses the workflow to generate rapid 3D models ... non-experienced interventional neurosurgeons" Improved this section. It is better to remove "From our knowledge, this study includes the highest number of cases (lines 150-151)" what does this sentence apport to the paper?
Response 4: Lines 175-179 – We deleted the sentence “From our knowledge, this study includes the highest number of cases” as you advised. The phrase “ This article discusses the workflow to generate rapid 3D models ... non-experienced interventional neurosurgeons” represents an introduction of all the points I have discussed in the next lines regarding the variables of the 3D printing process that can lead to an inaccurate model by taking into consideration other studies and their experience. Also, the advantages of using the solid model before choosing an endovascular approach were reviewed later in text, including the overall better agreement between the three interventional neurosurgeons with different levels of experience and the changes made in some cases regarding the treatment planning after analysing the 3D printed model.
We made some changes to improve the understanding of the sentence.
“This article summarizes the workflow to generate 3D models and points out some key elements for speeding the process while maintaining the accuracy of models and also demonstrates the advantages of having a solid 3D model before choosing between two different endovascular procedures for both experienced and less-experienced interventional neurosurgeons.”
Point 5: Lines 152-156: "Endovascular treatment of intracranial aneurysms requires a good assessment of the... 3D printing process". At this point can be useful to report and discuss the role of 3D printed models also in surgery to help the surgeon to avoid clips malpositioning, wrapping and revascularization. Please look at these references: Revascularization for complex intracranial aneurysms. Neurosurg Focus. 2008;24(2):E21. doi: 10.3171/FOC.2008.25.2.E21. -- Wrapping of intracranial aneurysms: Single-center series and systematic review of the literature. Br J Neurosurg. 2015;29(6):785-91. doi: 10.3109/02688697.2015.1071320. -- Comparison of 3D intraoperative digital subtraction angiography and intraoperative indocyanine green video angiography during intracranial aneurysm surgery. J Neurosurg. 2018 Jul 13;131(1):64-71.
Response 5: Lines 167-175: We made some changes to this paragraph and included a discussion regarding the usefulness of printed models in aneurysm surgery supported by articles published by Scerrati and colab and Kang et al.
“The physical models provide an overview of the complete anatomy of the aneurysm, the relationship with the arterial branching pattern, aspects that were found to be accurate in comparison to the intraoperative reality. Furthermore, they create a good training environment for surgeons to test surgical approaches, for clip selection and direction minimizing the risk of clip malpositioning. Scerati et al [15] found a high correspondence between the clipping strategy selected in preoperative planning and the strategy used during surgery. Yin Kang et al [16] in his study showed that 3d printed model will also provide the surgeon a good tactile experience and will help in adjusting the head position and choosing a shorter path to the aneurysm.
The articles published in the literature are focused mostly on surgical or surgical vs. endovascular treatment and simulation and don’t include a large sample of printed models.”
Point 6: Lines 182-156: ". In our study, we demonstrated a very good agreement between the 10 years experienced interventionist and 4 years experienced interventionist when they analyzed the data set that included the 3D printed model (Figure 6)" . add percentage data. - "Also, the agreement was higher between all physicians after they examined the printed model." All? How many physicians? add percentage data or removed.
Response 6: Lines 226-233: We added the value of kappa coefficient in parentheses and specified the classes that were correlated in the second part of the paragraph. “In our study, we demonstrated a very good agreement (k=0.85) between the 10 years experienced interventionist and 4 years experienced interventionist when they analyzed the data set that included the 3D printed model (Figure 6), a kappa coefficient higher than before seeing the solid model (k=0.7). Also, we found a good agreement between the 10 years experienced surgeon and the resident (k=0.7) and between the 4 year experienced surgeon and the resident (k=0.724) after they examined the printed model in comparison with the agreements before adding the 3D printed model to the data set (k=0.64 and, respectively k=0.48).”
We deleted the sentence “ Also, the agreement was higher between all physicians after they examined the printed model."
Point 7: Lines 220-221: "because they were relatively breakable after 3 days" why ? please discuss
Response 7: Lines 273-287: Our study used classic whole models, not hollow. We tried to make hollow models for endovascular simulation and training purposes. Unfortunately, we have to further investigate the printing workflow to obtain these models because they became breakable/fragile after about 3-5 days and a simulation with a catheter was not possible.
We modified the phrase and added reference a paper that discusses in detail the workflow of vascular model fabrication using different 3D printing technologies: “A direction of future research for our work is regarding the fabrication of hollow models for actual simulation of endovascular procedures and training. We tried to make hollow models for this purpose (Figure 7) but we need to further investigate the post-processing part of these printed models because they were relatively breakable after 3 days. So the short period of time may not give the chance for interventional surgeons to make an endovascular simulation. One explanation for model deterioration would be that we maybe didn’t use the right resin for fabrication because a hollow model must not have porosities and should be relatively flexible. Another reason is probably the fact that we didn’t expose the model after printing to UV light for curing purpose [31]. However, our classic whole 3D printed models remained intact even after 1 year.
Point 8: Figure 4 is useless in this way. Could the authors add more informations?
Response 8: Figure 4 is a graphical representation of distribution of aneurysms location in the Results section for a better visualization. We will remove it because the percentages are already written in the text. We renumbered the figures.
Reviewer 2 Report
The work of Marciuc and collaborators describes a novel and cutting-edge tool for intracranial aneurysms treatment. They designed and 3D-printed patient-specific models and analyzed through neurosurgeons' opinion.
Minor comments are required to implement the current manuscript:
1- The authors didn't specify any sterilization methods they planned to use before implanting the 3D-printed model
2- The authors should elaborate more about the parameters that the different groups of neurosurgeons took in consideration in order to analyze the models
3-Are the authors taking in consideration other kind of materials? Are they sure that the resin are using is biocompatible to be implanted?
4-Regarding the statistical analysis paragraph: the authors should add references in which the different Kappa values range are reported.
Author Response
Response to Reviewer 2 Comments
Point 1: The authors didn't specify any sterilization methods they planned to use before implanting the 3D-printed model
Response 1: The 3D printed models were not sterilized because they were not taken in the operating room. The neurosurgeons analyzed the models retrospectively along with the computed tomography angiography images and reconstructions and were asked to decide regarding the endovascular approach, whether they would have chosen direct coiling or stent-assisted coiling. However, there are several methods for sterilization (steam autoclave, ethylene oxide, gamma radiation) but there are also some aspects that must be taken into consideration and the most important is the deterioration of the model. The most recent and most appropriate method of sterilization is cold gas plasma but is a surface-based treatment that requires a good exposure of all surfaces.
Point 2: The authors should elaborate more about the parameters that the different groups of neurosurgeons took in consideration in order to analyze the models
Response 2: Lines 111-113 - We added the parameters the neurosurgeons took into consideration when they analyzed the printed models: “They noted the type of endovascular procedure they would have chosen (direct coil embolization or stent-assisted coil embolization) taking into consideration the shape, size, neck and orientation of the aneurysm, the parent vessel and whether there is an emerging vessel from the aneurysmal sac.” (lines 109-112).
Point 3: Are the authors taking in consideration other kind of materials? Are they sure that the resin are using is biocompatible to be implanted?
Response 3: In this study we didn’t used biocompatible resin because the models were retrospectively used for case analyses. There are also different types of biocompatible resin (class 1, class IIa, IIb and class III) and new directions of research are encouraged to determine the most suitable printing material used for implantation.
Point 4: Regarding the statistical analysis paragraph: the authors should add references in which the different Kappa values range are reported.
Response 4: Lines 120-122 - We added a reference data regarding the kappa coefficient in the Statistical analysis paragraph, as you advised. The reference was also included it in the bibliography section.
“Kappa values lie in a range of 0 to 1. Values between 0,8 – 1 represent a very good agreement or almost perfect; good agreement or substantial agreement between 0,6 – 0,8; moderate agreement between 0,4 – 0,6; poor or fair agreement between 0,2 – 0,4; slight agreement for values less than 0,2 and no agreement for values below 0 [9].
- Landis JR, Koch GG. The measurement of observer agreement for categorical data. Biometrics. 1977, 33(1), 159-74.
Round 2
Reviewer 1 Report
Authors solved all criticisms.
This manuscript is a resubmission of an earlier submission. The following is a list of the peer review reports and author responses from that submission.